# Cervical Spondylotic Myelopathy—Diagnostics and Clinimetrics

**DOI:** 10.3390/diagnostics14050556

**Published:** 2024-03-06

**Authors:** Józef Opara, Martyna Odzimek

**Affiliations:** 1Department of Physiotherapy, The Jerzy Kukuczka Academy of Physical Education, 40-065 Katowice, Poland; 2Doctoral School, The Jan Kochanowski University, Żeromskiego 5, 25-369 Kielce, Poland; 3Institute of Health Sciences, Collegium Medicum, The Jan Kochanowski University, al. IX Wieków Kielce 19A, 25-516 Kielce, Poland

**Keywords:** *cervical spondylotic myelopathy* (*CSM*), clinimetrics, degenerative cervical myelopathy, diagnostics, spinal cord compression

## Abstract

Cervical myelopathy is referred to in many ways in the English literature, for example, as *cervical spondylotic myelopathy* (*CSM)*, *spondylotic radiculomyelopathy* (*SRM*) or *degenerative cervical myelopathy* (*DCM*). In addition, more frequent occurrences are noted in older adults and to a greater extent in men. The causes of the effects of cervical myelopathy may be the appearance of lesions on the spinal cord, ischemia due to compression of the vertebral artery and repeated micro-injuries during maximal movements—hyperflexion or hyperextension. It is well known that lesions on the spinal cord may occur in a quarter of the population, and this problem is clearly noted in people over 60 years old. The symptoms of SCM develop insidiously, and their severity and side (unilateral or bilateral) are associated with the location and extent of spinal cord compression. Neurological examination most often diagnoses problems in the upper limbs (most often paresis with developing hand muscle atrophy), pyramidal paralysis in one or both lower limbs and disorders in the urinary system. To make a diagnosis of CSM, it is necessary to perform MRI and neurophysiological tests (such as EMG or sensory and/or motor-evoked potentials). The use of appropriately selected scales and specific tests in diagnostics is also crucial. This narrative review article describes the latest knowledge on the diagnosis and clinimetrics of cervical spondylotic myelopathy in adults and provides future directions.

## 1. Introduction

In the English literature, cervical myelopathy is described in various ways and there are many synonyms for this disorder. The most common terms include *cervical spondylotic myelopathy* (*CSM*), *spondylotic radiculomyelopathy* (*SRM*) and *degenerative cervical myelopathy* (*DCM*) [1,2,3,4]. Analyzed scientific research indicates that CSM is one of the most common causes of disability in older people [5], and other work shows that it can be classified as a civilization disease [6]. Research also shows that there are problems regarding lack of clarity in the appropriate diagnosis of patients, especially those over 60 years of age, and this problem may affect as many as 90% of this group [6]. However, Nouri et al., on behalf of a group of experts from AO Spine RECODE-DCM Research, presented crucial research results indicating that problems in the form of spinal cord compression may occur in up to a quarter of the population of healthy people, and this disorder is significantly more common in the age group over 60 years old (about 35% of people in this age group) [7]. Based on cases surgically treated in the Netherlands, it was estimated that the incidence of CSM in the local population is much lower [8] than according to studies conducted in Taiwan, where the problem occurred statistically four times more often per 100,000 inhabitants [9]. It is worth adding, however, that the analysis by scientists from Taiwan was based on many years of observation, especially of hospitalization rates. This study noted that cervical myelopathy disorders were more common in older age groups, and males were more likely to present with the problem [9]. It is worth noting that when spinal canal stenosis exceeds more than 30%, CSM may develop. There are numerous theories explaining the development of this disorder in the scientific literature, but some of the most influential are (1) direct pressure of degenerative changes (osteophytes, intervertebral disc or ligamentum flavum) on the tissues around the spinal cord or directly on the spinal cord; (2) abnormal blood supply, which is most often caused by pressure on the vertebral artery or its branches; and (3) frequent and overlapping micro-injuries that may occur during incorrect flexion and/or extension movements in the cervical spine. Some authors draw attention to the significant impact of the occurrence of CSM on the ossification of the posterior longitudinal ligament [10,11].

## 2. Diagnostics

Analyzing numerous scientific publications, the authors point out that increasing narrowing within the spinal canal may cause significant pressure on the nerve roots emerging from this level and, in the case of disease progression, on the pyramidal pathway. The pathogenesis of cervical spondylotic myelopathy may be caused by numerous factors, both statistical and dynamic, affecting the human body throughout its life. The first symptoms that patients notice, but do not necessarily consult a doctor about, include a pain in the cervical spine that is difficult to characterize, unpleasant sensations in the hand (tingling, numbness), a pain in the shoulder joint or shoulder area, frequent abnormalities and changes in gait pattern, and less common dysfunctions in the urogenital system (urinary incontinence, impotence). In the initial phase, the symptoms are unnoticeable to the patient and are largely ignored, which causes them to worsen and their functional status to deteriorate. The severity of CSM and lateral symptoms (unilateral or bilateral) is largely related to the location and degree of spinal cord compression. During the neurological examination of patients with CSM, the most common symptom is paresis, especially in the upper limbs, which may lead to muscle atrophy in the hand or both hands. Much later, problems in the lower limbs are noticed, especially in the form of pyramidal paresis. Patients with CSM are characterized by psychological problems and are often diagnosed with depression or mood disorders. During the detailed diagnosis of patients, numerous diseases and previously implemented elements of therapy are taken into account. It is worth noting that the examination of cerebrospinal fluid is negative, and the results of neuroradiological examinations are relevant. First of all, patients are sent for an MRI examination; then, neurophysiological tests are performed. One of the most important tests in the diagnosis of CSM is EMG testing, but sensory and/or motor-evoked potentials (*SEP* and *MEP*) are also measured [12]. The most characteristic images are T2-weighted and diffusion tensor (DTI) images in a nuclear resonance (MR) examination. Based on a prospective study which included a group of 60 patients with neurological symptoms suggestive of CSM and a control group of 30 healthy subjects, Mostafa et al. stated that DTI indices are valuable tools for the quantitative assessment of degenerative CSM in addition to routine cervical spine MR [13]. The disproportion between spinal compression and the clinical picture is striking. Significantly important research on predictive factors in the course of non-myelopathic degenerative cervical cord compression was conducted by a team of researchers led by Kadanka. The above observational study demonstrated that the development of degenerative cervical myelopathy was largely associated with numerous changes in MRI examinations (changes in the diameter of the cervical cord and cervical canal, changes within the cross-section) and electrophysiological (EMG, SEP, MEP) and clinical observations such as such as changes in gait and its execution time [14]. Moreover, scientific research suggests that the course of CSM may occur in multiple ways and be a variable phenomenon. These studies confirm the above theories that there are changes in signals in MRI examinations and in electrophysiological examinations (in particular, MEP and EMG) which constitute an important and very valuable complement to the patient’s diagnosis. Hypotension on T1 MRI images should be considered a sign of more advanced disease [15]. It is worth noting that the prevalence of cervical spine pain was estimated to be approximately 34%, and less than half of this group met the criteria for chronic pain, i.e., pain where symptoms last longer than six months. Additionally, researchers point out that symptoms lasting longer than four weeks are reported more often in women than in men [16].

## 3. Clinimetrics

As with many diseases affecting humans, it is difficult to predict its course, but cervical spondylotic myelopathy is most often slowly progressive, characterized by the gradual intensification of symptoms [7]. Surgical treatment is important and is necessary in some patients, but due to the initial improvement, is often put off. The disease is largely progressive and symptoms rarely improve. The basis for the diagnosis of CSM includes the results of neurological examination, magnetic resonance imaging (MRI) and electrophysiological tests. Additionally, numerous scales and clinimetric tests are used to help make the final diagnosis and propose treatment methods [1,11]. The first clinimetric scales in neurology were stroke scales. The term ‘clinimetrics’ was first used in 1982 by Feinstein [17], and in neurology, it was employed in 1987 by Asplund to refer to strokes [18]. Professor Asplund was one of the authors of the Scandinavian Stroke Scale [19]. Clinimetrics means the measurement of clinical phenomena occurring in the patient. This specific field in medical research focuses on the creation and detailed evaluation of clinical indicators. Moreover, it contains many methods that are related to numerous fields of science. The diagnosis of cervical spondylotic myelopathy is too rarely made, and it is highly probable that the prevalence of CSM is underestimated. Patients with CSM often live with diagnoses of Guillain–Barré syndrome, spinal ischemia, amyotrophic lateral sclerosis, syringomyelia, spinal cord degeneration, tumors, multiple sclerosis and many other diseases or disorders [11]. There are several clinimetric scales used for the clinical assessment of patients with CSM [20]. The most commonly used scale is the Neck Disability Index (NDI), which is a simplified version of the Oswestry Low Back Pain Index [21]. It is a scale consisting of ten questions for which the respondent must choose one of six possible answer options. During the analysis, we collect information on the intensity of pain, but also its impact on everyday activities, such as recreation, sleep disorders, professional work or lifting objects. It is a useful and frequently used tool that has been translated into many languages and has been the basis for many scientific publications [11,22]. Additionally, the scale created in 1972 by Nurick (Classification System for CSM), which assesses in detail the degree of gait disorders, may be useful in the overall assessment of patients. This is one of the scales largely used by specialists in neurology, neurosurgery and orthopedics. On a six-point scale, the lowest value (grade zero) means there are no symptoms of the disease in the spinal cord, while grade five indicates a dependent patient who requires the care of others and uses a wheelchair or is unable to move [23]. A method with a lesser role in the diagnosis of patients with CSM is the use of the Myelopathy Disability Index (MDI), which was created in 1996 by Casey and co-creators, who halved the standards contained in the Stanford Health Assessment Questionnaire (HAQ). There are four possible answers to each question, thanks to which it is possible to determine the severity of the disorder. The questions mainly concern getting up, washing or performing everyday activities. The scale is largely used by doctors, especially rheumatologists [23]. The Japanese Orthopedic Association score (JOA score) is a scale created to assess a patient’s neurological status. In addition, it allows the doctor to re-examine and assess the therapeutic effects achieved. Slightly different from the above scales, it assesses in detail the functioning of the body, especially in the scope of upper limb functions, lower limb functions (including, in detail, the ability to walk), urinary and digestive system functions and sensory functions [24]. The creation of the European Myelopathy Scale (EMS) in 1994 had important significance in the diagnosis of cervical myelopathy. It supplemented the JOA scoring with examination of the upper and lower motor neurons and also focused on detailed diagnostics in the area of the posterior roots and posterior columns [11,25,26]. A consequential, simple and reliable tool that cannot be omitted in this work is Rapid Hand Flick Time (RHFT). It allows you to assess the improvement in upper limb function after surgery due to CSM. This test is carried out in three time trials, and the patient performs a number of repetitions of closing and opening the hand as determined by the therapist [27]. The Triangle Step Test (TST) performed for examination of the lower limbs is equally simple. The test is performed in a sitting position and movements are performed in the shape of an equilateral triangle in a unit of time specified by the therapist [28]. It is also worth adding that in the case of CSM, the Ranawat scale and the Cooper scale were also used, but now they are used much less often [22].

## 4. Rare Cases of Cervical Myelopathy

Cases of the coexistence of two diseases in one patient have been reported: CSM with multiple sclerosis and transverse myelitis, or CSM with amyotrophic lateral sclerosis [29,30]. There have been cases where the first symptom of CSM was a non-traumatic, spontaneous epidural hematoma [31]. Avilés-Hernández et al. described nine cases of CSM after surfing—an ischemic patho-mechanism was assumed [32]. Some authors draw attention to the presence of CSM symptoms in cases of involuntary movements of the cervical spine. In addition, scientific research found a case report of a boy with Tourette’s syndrome who developed quadriparesis during CSM—significant improvement was observed after surgical treatment [33]. Mondal and Giri described two cases of athetotic cerebral palsy, in which involuntary movements led to symptoms of CSM with compression fractures of the first and second cervical vertebrae. In both cases, hyperventilation led to type 2 respiratory failure [34].

## 5. Discussion

The clinical symptoms of cervical spondylotic myelopathy (CSM) may be caused by compression and degenerative changes in the spinal cord, abnormal or significantly disturbed blood supply caused by compression of the vertebral artery, and micro-injuries during maximum flexion or in the cervical spine. In light of the latest scientific research, the incidence of CSM is significantly increasing, which may be due to the aging of the population [6,9,11]. Research conducted by Wu and co-authors showed that among patients hospitalized for CSM, the incidence of spinal cord injuries was significantly lower in the group of people undergoing surgical treatment than in the group of people receiving conservative treatment [9]. One of the latest studies presented by Khan and co-authors presented the results of a comprehensive literature search for 2010–2022 according to the PRISMA guidelines. The analysis took into account important demographic variables, disease severity and the method and characteristics of magnetic resonance imaging (MRI) in people with CSM. A total of 47 studies were included in the analysis, of which the highest potential was indicated in metric fractional anisotropy. Some studies in the above analysis demonstrated the great usefulness of MRI in the diagnosis of CSM, but the authors pay special attention to the fact that other modal configurations can also be very useful in detailed diagnostics. This is one of the newest methods and is still in the development phase; therefore, researchers emphasize that it is important to conduct further research into the use of individual MRI elements in the spinal cord in the course of CSM [35]. In another recent report from December 2023, Atchut et al. confirmed the role of diffusion tensor imaging (DTI) in the diagnosis of spinal stenosis. Additionally, the researchers noted that routine MRI may not be sensitive enough to detect small changes in the spinal cord. However, it is worth adding that the use of MRI diffusion tensor imaging (DTI) can have a significant function in detecting even small changes. In their work, the authors conducted research on a group of sixty-four people with spinal cord problems who met all the required inclusion criteria and provided written consent. Each study participant underwent routine magnetic resonance imaging, diffusion tensor imaging and other relevant indicators. The researchers noticed a statistically significant difference between the groups of people with and without spinal stenosis. People from the control group had higher scores in the Fractional Anisotropy (FA) variables and lower scores in the Apparent Diffusion Coefficient (ADC) group. Moreover, statistically significant differences were noted depending on the age of the respondents, where in people with stenosis, the average age was over 40 years. The authors concluded that the DTI, FA and ADC parameters may be important in detecting small changes in the spinal cord. Moreover, it is important to standardize the protocol that will be used in the diagnosis of people with CSM [36]. Also, recently, at the turn of November and December 2023, the team led by Haynes presented information on functional magnetic resonance imaging (fMRI) of the spinal cord and its use in clinical trials, including in CSM. The authors of the study showed that various forms of methodology and numerous spinal cord stimulation protocols were used during the analysis, which may contribute to the further development of this field of science. A detailed review of the literature on fMRI was conducted, including studies that included spinal cord examination. The authors of the study paid particular attention to the great technical difficulties in designing the research, the introduction of modern technologies and the discovery of new, previously unknown information about the spinal cord. Moreover, it is worth adding that examination of the spinal cord, especially when diagnosing CSM, may contribute to shortening the time of diagnosis and treatment. The authors emphasize that innovative research in the field of fMRI may contribute to the creation of new forms of patient treatment and the implementation of innovative diagnostic protocols [37]. In October 2023, Khan’s team and co-inventors presented a review on functional changes in the human brain in CSM patients using fMRI. The authors paid particular attention to the fact that the greatest changes were visible in the areas of the brain responsible for motor control. Moreover, significant changes were noticed in the superior frontal gyrus and occipital cortex, respectively. It is worth adding that the researchers presented information showing that there is a relationship between changes occurring in the brains of people with CSM and clinical indicators. However, this evidence is on a low or very low level because sufficiently large treatment and control groups are still missing. Nevertheless, it is worth noting that fMRI may be one of the elements facilitating the diagnosis of CSM due to the quantitative examination of changes in the brain. The field is still in the development stage [38]. The research conducted by Chang et al. involved an analysis that included patients with CSM and healthy people who underwent fMRI and structural MRI. Their study results showed that CSM patients presented a reduced fractional amplitude of low-frequency fluctuations (fALFF) scores in various locations in the brain and spinal cord. Moreover, the analysis showed that in patients with CSM, there is a reduction in the relationship between the thalamus and the cerebral cortex due to compression within the spinal cord. The above-mentioned changes lead to changes in the structure and functioning of the cerebral cortex. The obtained study results may contribute to the development of the field related to non-imaging of the brain and spinal cord in people with CSM [39]. Many experts are of the opinion that the assessment of quality of life (QoL), a kind of psychometrics, belongs in a broader sense to clinimetrics. The study designed by Oh and co-authors assessed the quality of life of people with CSM and other chronic diseases. The analysis was performed using the 36-Health Survey (SF-36). People who were included in the study had to meet the criteria specified by the creators and sign written consent to participate in the study. The researchers did not note any statistically significant differences in age groups, but people from the study group had greater disability. The analyses conducted show that CSM may have a significant impact on the quality of life of patients and their daily functioning [40].

## 6. Future Directions

Attempts to chart the future directions of research in CSM are few in the literature. First was Fehlings et al., who prepared a narrative overview introduction for the Special Issue of *Spine* dedicated to CSM. The authors of the above study decided to summarize the current state of knowledge and determine the next steps that may be taken in future research. It is worth emphasizing that, in their opinion, it is necessary to unify the CSM naming system, complete it in detail and unify it. Additionally, it is worth mentioning that the authors postulate that ossification of the posterior longitudinal ligament (OPLL) should be included in the nomenclature. The authors point out that the above steps may contribute to the development of the field and involve numerous research teams in research. It should be added that it is important to introduce research specifically on the incidence of CSM, especially in terms of degenerative changes, as well as the occurrence of OPLL in European and non-European populations. It is necessary to create or improve CSM diagnostic methods to standardize the obtained results and enable scientists to exchange experiences and introduce new treatment methods. Moreover, the introduction of new pharmacological treatment methods may support surgical methods and reduce the effects of the disease. Further research with larger studies and control groups is needed to implement preventive methods and increase the awareness of sick people and their families [41]. In 2015, Nouri and co-authors proposed a new, more appropriate term, ‘Degenerative Cervical Myelopathy’ (DCM), which covers both CSM and ossification of the posterior longitudinal ligament (OPLL). The authors of the study believe that the degenerative nature of the disease may be related to the older age of the patients [42]. However, in a 2020 review article, the same author claimed that there has been significant scientific progress in the last decade when it comes to CSM and DCM. All this is thanks to high-quality research, the main goal of which is to standardize the nomenclature of the above phenomena. However, the above studies show negative effects, with significant disability in patients after surgery. According to the authors, it is important that future research focuses on effective surgical treatment and the introduction of individual treatment plans that will enable the fastest return to a functional state. It is worth emphasizing that the development of methods for imaging and conducting CSM diagnostics is important. Additionally, there is promise in research and advanced techniques in the field of MRI, which may improve the standards of care for people with CSM [43]. According to a review article by Haynes and co-authors, it can be concluded that further research in the field of fMRI of the spinal cord will contribute to the development of treatment for people with diseases in this area and will constitute the basis for implementing new assessment methods and developing treatment strategies [37]. However, the article by Khan and co-authors on the same topic gives great hope that the implementation of fMRI may facilitate the diagnosis of CSM. This study highlights even small changes in the brain, especially those responsible for motor control [38]. It is worth adding that the research of Chang and co-authors gives great hope that neuroimaging may contribute to significant development in the diagnosis of CSM [39].

In 2022, Yanez Touzet, in the name of the AO Spine RECODE-DCM Steering Committee, published a systematic review to establish a core measurement set on clinical outcome measures and their evidence base in DCM. The authors identified 29 outcome instruments from 52 studies published between 1999 and 2020. They measured neuromuscular function (sixteen instruments), life impact (five instruments), pain (five instruments) and radiological scoring (five instruments). No instrument had evaluations for all 10 measurement properties, and <50% had assessments for all three domains (i.e., reliability, validity and responsiveness). There was a paucity of high-quality evidence. There were no studies that reported on structural validity and there was no high-quality evidence that discussed content validity. The authors identified nine instruments that are interpretable by clinicians: the arm and neck pain scores; the 12-item and 36-item short-form health surveys; the Japanese Orthopaedic Association (JOA) score; the modified JOA and JOA Cervical Myelopathy Evaluation Questionnaire; the neck disability index; and the visual analogue scale for pain. These include six scores with barriers to application and one score with insufficient criteria and construct validity. This review aggregates studies evaluating outcome measures used to assess patients with DCM. But the authors conclude that there is a need for a set of agreed-upon tools to measure outcomes in DCM. These findings will be used to inform the development of a core measurement set as part of AO Spine RECODE-DCM [44].

Recently (June 2023), Costa and 34 other neurosurgeons published detailed recommendations on behalf of the Italian Neurosurgical Society (SINch) which were worked out using the Delphi method. They divided this document into seven chapters: 1. Natural course and clinical presentation. 2. Diagnostic tests. 3. Conservative treatment vs. surgical treatment. 4. Anterior surgical treatment. 5. Posterior and combined surgical treatment. 6. Role of neurophysiological monitoring. 7. Follow-up and outcome. As functional measures, the experts recommend the modified Japanese Orthopedic Association scale (mJOA), Nurick grade and myelopathy disability index [45].

As for the future of clinimetrics, in our opinion, the future will see the use of modifications, i.e., shorter versions, of some well-known scales. The purpose of these modifications is to reduce the time needed to complete surveys. Moreover, regarding the assessment of quality of life, perhaps soon, the QoL questionnaire will be specific to CSM. This devastating syndrome is one of the most common causes of disorders in the spinal cord area. Additionally, it is worth noting that it may be reasonable to expect that the incidence of this disease will increase as the population ages. According to the latest (10 February 2024) preprint information by Yanez Touzet et al., in the future, smartphone apps will be used to objectively monitor performance outcomes in DCM [46].

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
