# Peer review of "Cervical Spondylotic Myelopathy—Diagnostics and Clinimetrics"

_diagnostics, 2024, doi:10.3390/diagnostics14050556_

Round 1

Reviewer 1 Report

Comments and Suggestions for Authors

1.    A latest paper mentions Outcome measures used for CSM. As functional measures they recommend modified Japanese Orthopedic Association scale (mJOA), Nurick grade and myelopathy disability index.

Costa F, Anania CD, Agrillo U, Roberto A, Claudio B, Simona B, Daniele B, Carlo B, Barbara C, Ardico C, Battista CG. Cervical Spondylotic Myelopathy: From the World Federation of Neurosurgical Societies (WFNS) to the Italian Neurosurgical Society (SINch) Recommendations. Neurospine. 2023 Jun;20(2):415.

2.    Adding tables to mention the various clinimetric scales and diagnostic modalities will make this a better review.

3.    In each diagnostic technique advantages disadvantages can be listed with most commonly used test.

4.    Similarly, for clinimetric scales/indexes they can be listed and information can be summarised to know which is still in use and which has now been discontinued or is no longer considered useful.

5.    A latest article talks about role of magnetic resonance diffusion tensor imaging in the accurate evaluation of cervical spondylotic myelopathy.

Mostafa NS, Hasanin OA, Al Yamani Moqbel EA, Nagy HA. Diagnostic value of magnetic resonance diffusion tensor imaging in evaluation of cervical spondylotic myelopathy. Egyptian Journal of Radiology and Nuclear Medicine. 2023 Oct 16;54(1):175.

Author Response

      Thank you very much for your valuable comments and suggestions! Two latest papers mentioned by you (Costa F et al. Neurospine. 2023 Jun;20(2):415, and Mostafa NS et al. Egyptian Journal of Radiology and Nuclear Medicine. 2023 Oct 16;54(1):175) have been added to the list of references. 

As for various clinimetric scales: now we had more time and found the most valuable review paper: Yanez Touzet A. et al. Clinical outcome measures and their evidence base in degenerative cervical myelopathy: a systematic review to inform a core measurement set (AO Spine RECODE-DCM), and described in our manuscript. Yanez Touzet in the name of the AO Spine RECODE-DCM Steering Committee published a systematic review to inform a core measurement set on clinical outcome measures and their evidence base in DCM. The authors identified 29 outcome instruments from 52 studies published between 1999 and 2020. They measured neuromuscular function (16 instruments), life impact (five instruments), pain (five instruments), and radiological scoring (five instruments). No instrument had evaluations for all 10 measurement properties and <50% had assessments for all three domains (i.e., reliability, validity, and responsiveness). There was a paucity of high-quality evidence. There were no studies that reported on structural validity and no high-quality evidence that discussed content validity. The authors identified nine instruments that are interpretable by clinicians: the arm and neck pain scores; the 12-item and 36-item short-form health surveys; the Japanese Orthopaedic Association (JOA) score, modified JOA and JOA Cervical Myelopathy Evaluation Questionnaire; the neck disability index; and the visual analog scale for pain. These include six scores with barriers to application and one score with insufficient criterion and construct validity. This review aggregates studies evaluating outcome measures used to assess patients with DCM. However, the authors conclude that there is a need for a set of agreed tools to measure outcomes in DCM. These findings will inform the development of a core measurement set as part of AO Spine RECODE-DCM.

This should satisfy any researcher with a detailed interest in clinimetrics in CSM.

Reviewer 2 Report

Comments and Suggestions for Authors

The paper reviewed is aimed to present the latest knowledge on the diagnosis and clinimetrics of cervical spondylotic myelopathy and provides future directions.
The paper is submitted for the special issue of the journal focused on Diagnostics, Treatment, and Rehabilitation of the Spinal Cord Injury. This fact that supposed that the issue is aimed to present to the readers, not only the most recent original papers, but also to refresh the basic knowledge, and to give the perspective from the classical interpretation of the problem to the future directions. This point of view is equally feasible for the problem of cervical myelopathy, which has long-lasting history and some long discussion questions. The current paper gives general information regarding interpretation of the term “myelopathy”, especially under the light of the current methods of visualization. The authors present and discuss some unsolved problems of the phenomenon of spinal cord compression, including possible underlying mechanisms and the diagnostic approaches as well. The main focus of he paper is in regard of clinimetrics – development, practical application and critical interpretation of the scales and assessment instruments in the field of cervical myelopathy.
The paper makes positive impression in terms of structure, content and narration of the material. Most of the literature sources cited are from the recent publications or classical studies, which are actual for the concetualising of the problem discussed. Despite some limitations, first, non-systematic approach to literature search and analysis, the paper is interesting to read, and as a part of the special issue can definitely contribute to the better understanding of the problem.
From the reviewer’s point of expertise and taking into account the chapter dedicated to the rare observations it is important also to include some information regarding cervical myelopathy in children with skeletal dysplasias and metabolic inborn errors like mucopolysaccharidoses, Larsen syndrome, diastrophic dysplasia, metatropic dysplasia and so on, which is important to keep in mind, especially taking into account the growing survival rate of patients with above mentioned conditions in presence of emerging targeted therapies, and growing number of publications in surgical strategies of prevention and management of cervical spine compression in children with these disorders.

Author Response

Note to Reviewer 2 Comments and Suggestions for Authors            

Thank you very much for your positive review!

Especially we thank you for your positive impression in terms of structure, content, and narration of the material.

As for your suggestion about some information regarding cervical myelopathy in children with skeletal dysplasias and metabolic inborn errors like mucopolysaccharidoses, Larsen syndrome, diastrophic dysplasia, metatropic dysplasia and so on: I'm sorry that pediatrics is outside my area of expertise skill. That`s why we decided to add in the abstract that this is a review about CSM in adults.

Obviously, this is a very interesting topic, worth some separate review paper.

Reviewer 3 Report

Comments and Suggestions for Authors

I would like to congratulate the authors for their work. The purpose of the manuscript is to present the diagnostics and clinimetrics regarding cervical spondylotic myelopathy. These aspects of disease were described in details and easily readable. This manuscript achieves its main purpose as a review, which is to provide up to date and concentrated information.  Also, the references are relative to the subject. Given the frequency and occurrence of cervical spondylotic myelopathy, I believe that this article will be useful for the involved surgical community.

Recommendations

-minor grammatical and editing corrections

-some figures depicting the main radiographic findings of the disease will increase the readability of the article

-beside the future directions section, a conclusion section is lacking from the article

Comments on the Quality of English Language

minor grammatical and editing corrections

Author Response

no reviewer 3
